genetics

Drosophila, physical fitness, climbing, exercise, DGRP, GWAS

**Author for correspondence:**
Nicole C. Riddle
e-mail: riddlenc@uab.edu

# Exercise-induced changes in climbing performance

Louis P. Watanabe and Nicole C. Riddle

Department of Biology, The University of Alabama at Birmingham, CH464, 1720 2nd Ave South, Birmingham, AL 35294, US

NCR, 0000-0003-1827-9145

Exercise is recommended to promote health and prevent a range of diseases. However, how exercise precipitates these benefits is unclear, nor do we understand why exercise responses differ so widely between individuals. We investigate how climbing ability in *Drosophila melanogaster* changes in response to an exercise treatment. We find extensive variation in baseline climbing ability and exercise-induced changes ranging from −13% to +20% in climbing ability. Climbing ability, and its exercise-induced change, is sex- and genotype-dependent. GWASs implicate 'cell–cell signalling' genes in the control of climbing ability. We also find that animal activity does not predict climbing ability and that the exercise-induced climbing ability change cannot be predicted from the activity level induced by the exercise treatment. These results provide promising new avenues for further research into the molecular pathways controlling climbing activity and illustrate the complexities involved in trying to predict individual responses to exercise.

## 1. Introduction

Public health entities, including the World Health Organization and the US Centers for Disease Control, recommend regular exercise as an important part of healthy life [1,2]. Health benefits of regular exercise are diverse and range from improved cardiovascular performance and metabolic functions to improved mental well-being [3]. Regular exercise is recommended as a preventative measure for a variety of disease conditions and to promote healthy ageing [4–6]. In addition, exercise is a recommended treatment to alleviate symptoms for medical conditions ranging from metabolic to neurological and psychiatric diseases (for example, see [7–9]). Despite these countless benefits of exercise, many individuals do not engage in the recommended amount of regular exercise. For example, in the United States, only 53.3% of adults meet the guidelines of aerobic physical activity [1]. As exercise is one of the safest and most cost-efficient means to prevent health problems, increasing the population of individuals that participate in exercise is essential.

Health benefits of exercise in general are well documented, but individual responses to exercise treatments are immensely variable [10–13]. While a diverse set of exercise response variables (e.g. weight, blood pressure, insulin sensitivity, VO$_2$ max, etc.) are of interest, a wide range of responses are seen in individuals experiencing an exercise treatment for all of them. Frequently, individuals show the expected response to an exercise treatment, but the degree of this response differs considerably between individuals. However, there is typically a significant fraction of study participants, often up to 20%, that do not experience the expected response, and either show no change in response to the treatment, or actually show the opposite response. These non-responders in exercise studies have led to the concept of 'exercise resistance', which is controversial and likely dependent on the specific exercise treatment [14–16]. Unfortunately, we currently do not understand what causes the variability in exercise response, and it is impossible to predict which exercise treatment will work for which person.

Model systems like *Drosophila melanogaster* or *C. elegans* provide unique opportunities to investigate the sources of variation in exercise response. Likely, exercise responses are impacted by genetic background as well as a range of environmental factors, including diet, timing of the exercise treatment relative to circadian rhythms or drug treatments. To understand the impact of these factors and interactions between them, it is necessary to test many individuals in a large number of conditions, something non-mammalian model systems are ideally suited to. In *C. elegans*, swim training is used to exercise animals [17,18], and in *D. melanogaster*, several different exercise systems exist that all exploit the animals' natural geotaxis behaviour, their tendency to move to the top of an enclosure [19–22]. In both model systems, genetic tools exist to efficiently screen large numbers of target genes through, for example, RNAi knockdown collections, or to identify genes linked to exercise phenotypes using mapping populations. In addition, environmental conditions can be controlled and relevant behaviours and factors monitored in these model systems. These resources are essential to investigate the anticipated complex interactions between exercise, genotype and environment.

In Drosophila, rotation of vials is a well-established means for the induction of exercise. Several groups have developed rotational exercise devices independently, including the SwingBoat [23], the TreadWheel [24], the rotational exercise machine developed by the Key Laboratory of Physical Fitness and Exercise Rehabilitation of Hunan Province [25] and the Flygometer [26]. With these devices, rotational exercise has been shown to change metabolite levels, impact gene expression, precipitate weight changes, alter climbing ability, prevent age-related decline in physical fitness and more [24,25,27–29]. Our laboratory has used the TreadWheel and the REQS, a design based on the Treadwheel that allows us to monitor animal activity during exercise, to investigate the impacts of rotational exercise stimulation [30]. We have shown that exercise induction with the TreadWheel leads to changes in activity levels in a sex- and genotype-dependent manner [31]. A 5-day exercise treatment with the TreadWheel induces changes in animal weight, again in a sex- and genotype-dependent manner [32]. Thus, the Drosophila system is ideally suited to investigate the factors contributing to the variation in exercise response.

Here, we use a subset of the Drosophila genetic reference panel (DGRP), a collection of wild-derived inbred lines for genetic mapping, to study exercise-induced responses in climbing ability. We administered a 5-day exercise treatment with the TreadWheel and investigated the genetic pathways controlling climbing ability and exercise-induced changes in climbing ability. We found that climbing ability as well as the change in climbing ability precipitated by an exercise treatment is highly variable in this strain collection and that both traits strongly depend on sex and genotype. Genome-wide association studies identify several strong novel candidate genes impacting climbing activity, with genes functioning in cell–cell signalling and the regulation of multi-organ processes over-represented among the candidate genes. Interestingly, the response to the exercise treatment in terms of climbing ability is not related to the amount of exercise being performed, and there is no relationship between the responses seen in males and females of the same genotype. Together, our results illustrate the complex interactions between genetic background, sex and exercise that determine exercise responses such as the change in climbing ability measured here.

# 2. Results

## 2.1. Variation in climbing ability in the DGRP

To get a better understanding of the impact of exercise on physical fitness, specifically climbing ability, in *D. melanogaster*, we used a subset of DGRP lines. Three- to 5-day-old individuals from these lines were subjected to a 5-day exercise treatment, exercising in response to rotational stimulation on the

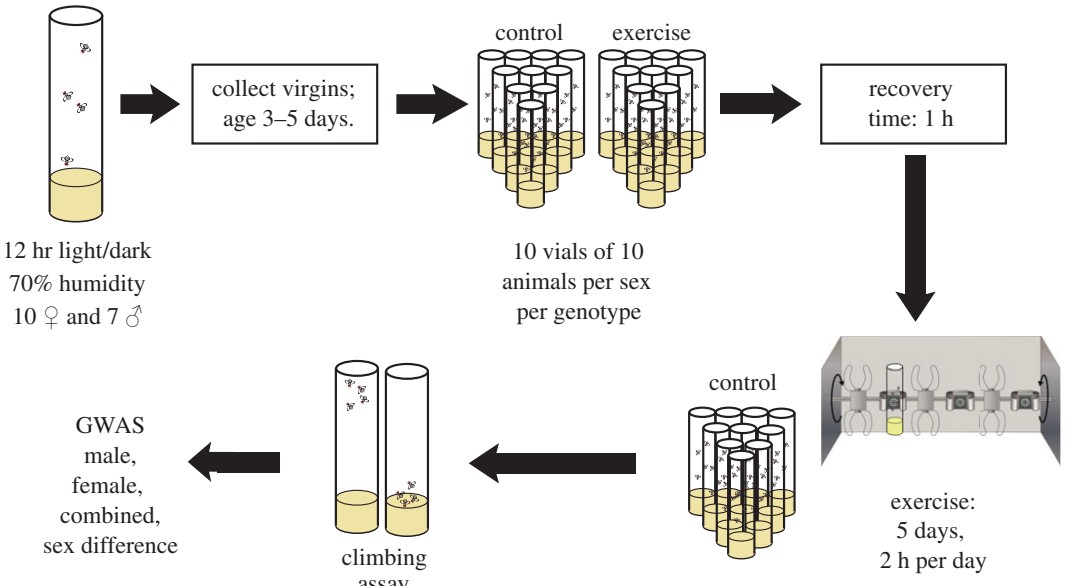

**Figure 1.** Experimental set-up. Flow diagram illustrating how animals were collected, exercise-treated and assayed for climbing ability.

Treadwheel for 2 h each day (figure 1 for details). Control animals were handled in the same fashion but did not receive an exercise treatment. On day 6, physical fitness in the exercise-treated and control animals was assayed using a rapid iterative negative geotaxis (RING) assay to determine a climbing index for each fly line. With this measure, a larger climbing index indicates that the flies were able to climb the vial walls faster after having been knocked to the bottom of the enclosure (range of possible climbing indexes: 1–4). We find that climbing ability, as measured by the climbing index, differs significantly between the DGRP lines, both in the treated and in the control animals ($p < 2.2 \times 10^{-16}$, Kruskal–Wallis test, figure 2). Climbing ability also differs significantly between the sexes in both treated and control animals ($p < 2.2 \times 10^{-16}$, Kruskal–Wallis test), with males typically showing higher climbing indexes and thus increased climbing ability compared to the females. In males, the climbing index ranges from 1.17 (controls and treated animals, line 359) to 2.05 in control (line 358) and 1.99 in treated animals (line 774). In females, the climbing index ranges from a low of 1.01 in controls and 1.05 in treated animals (both line 639) to a high of 1.80 in controls (line 315) and 1.87 in treated animals (line 852). While in females, the maximum and minimum per line climbing indexes in the treated animals are higher than in the control animals, there is no general treatment effect, either in males or in females ($p = 0.1854$ and $p = 0.2306$, Kruskal–Wallis test). Together, the data demonstrate that there is significant variability in climbing ability as measured by climbing assays among the DGRP lines.

To further examine the variation in climbing indexes in our study population, we used a mixed linear model to partition variance. We started with a model that included the fixed effects of sex, genotype, treatment and their interactions. In addition, we included two random effects, the position of the vial in the assay apparatus as well as the vial number. Finally, our initial model included 'picture' as a fixed variable, as the climbing assay was performed six times per set of animals, resulting in six digital images. Using the Akaike information criterion (AIC), we compared various models using a 'drop-1-out' approach. The model with the best fit included the factors sex, genotype, treatment, vial position and number, as well as the interactions between sex and genotype and sex and treatment, all of which contributed significantly to the model (for additional details, see Material and methods). This model was slightly better than a model that included all interactions between sex, genotype and treatment (AIC −1774.1 versus −1773.3, $p = 3.77 \times 10^{-6}$, Chi-square). These analyses highlighted the impact of interaction effects on climbing ability, illustrating the complex factors impacting this phenotype.

Next, we investigated the relationship between the climbing ability of animals of the two sexes and in the two exercise conditions. Examining first the relationship of climbing ability between the control and exercise-treated animals, we find that in both males and females, there is a high correlation between the climbing indexes between animals from the same DGRP line (electronic supplementary material, figure S1). In females, the correlation is 0.905 (Pearson's product-moment correlation, $p = 2.194 \times 10^{-}$

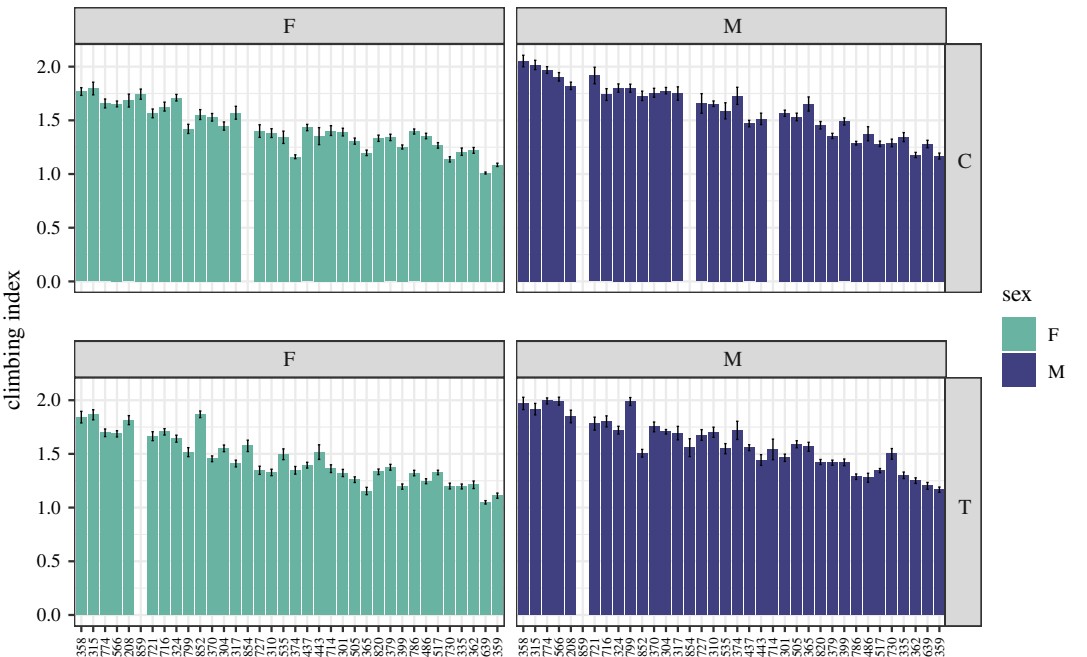

**Figure 2.** There is extensive variation in climbing ability among this set of DGRP strains. Mean climbing indexes (+/− standard error of the mean) are plotted on the Y-axis for each of the DGRP lines included in this study (X-axis). A larger climbing index indicates that the animals were able to climb the vial walls faster after having been knocked to the bottom of the enclosure. The data are shown separated by sex (F—female, M—male) and treatment (C–control, T–exercise-treated). Sample size: approximately five replicate vials with six repeated measures each per sex/genotype/treatment combination.

[13]), while in males, the correlation is 0.933 ($p = 2.809 \times 10^{-15}$). Comparing the climbing indexes between males and females of the same DGRP lines, the correlations are also high, with control animals showing a correlation of 0.821 between the sexes ($p = 4.963 \times 10^{-9}$), and exercise-treated animals showing a somewhat lower correlation of 0.744 ($p = 3.041 \times 10^{-7}$). Together, these data suggest that the animal's genotype strongly impacts climbing ability, irrespective of the sex of the animal and the treatment applied.

## 2.2. The relationship between climbing ability, animal activity, weight, and lifespan

Next, we asked how climbing ability in the DGRP population relates to the animal activity of the strains that we have measured previously [31], focusing on the data from the unexercised control animals. In females, we find a negative correlation between climbing index and basal activity levels (Pearson's product-moment correlation $r = -0.377$, $p = 0.04767$), meaning that animals with a low baseline level of activity tended to have a high climbing index, while animals with high baseline activity levels tended to have lower climbing indexes (electronic supplementary material, figure S2A). However, in males, there is no correlation evident between climbing index measures and basal activity levels (Pearson's product-moment correlation $r = 0.068$, $p = 0.7404$; electronic supplementary material, figure S2B). Similarly, there is no significant correlation between exercise activity levels and climbing index in either sex (Pearson's product-moment correlation $r = 0.203$ (females) and $r = 0.205$ (males), $p = 0.299$ and $p = 0.305$, electronic supplementary material, figure S2C&D). We have also carried out these analyses with the data from the animals that completed the 5-day exercise treatment and, again, find no significant correlation between climbing index and activity level (Pearson's product-moment correlation $r = -0.279$–$0.146$, $p > 0.15$). We do, however, find a positive correlation between the climbing indexes measured here and the startle response measured by Mackay and colleagues (Pearson's product-moment correlation $r = 0.373$ (F) and $0.442$ (M); $p = 0.03234$ (F) and $0.01005$ (M)) [33]. Together, these data suggest that there is no clear relationship between the amount of activity animals from different fly strains engage in and their overall fitness as measured by a climbing assay. Instead, it appears that the climbing assay provides a measure more similar to the startle assay, possibly providing a combined assessment of both the strength of the startle response to disturbance and physical fitness, i.e. climbing speed.

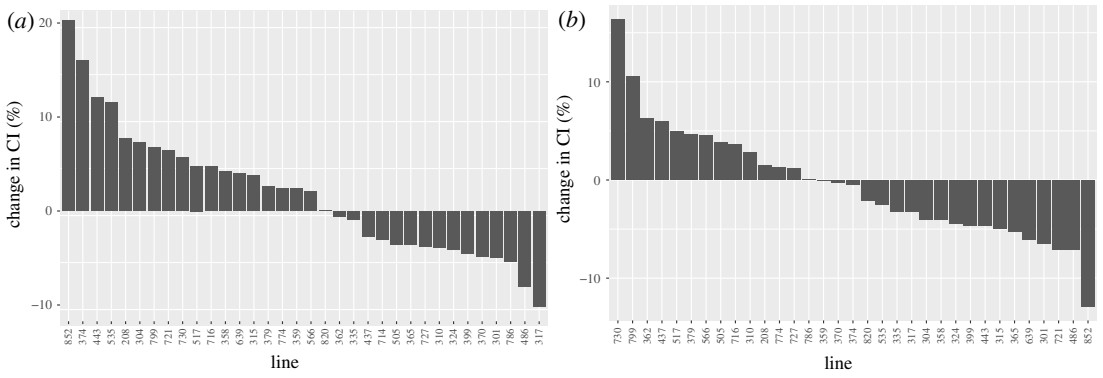

**Figure 3.** The impact of exercise on climbing ability differs significantly between the DGRP lines. The change in climbing index [(mean climbing index of the exercise-treated animals − mean climbing index of the control animals)/mean climbing index of the control animals *100] is plotted on the Y-axis for each of the DGRP lines included in this study (X-axis). A positive change indicates that the exercise treatment led to an increase in climbing index, while a negative change indicates that the treatment led to a decrease in climbing index. (a) Data from females. (b) Data from males.

We also investigated the relationship between climbing ability and two additional characteristics, lifespan and weight. Typically, physical fitness is associated with longer lifespans [34,35]. If this relationship exists in Drosophila, we expect that strains with higher climbing indexes would show longer lifespans. Using the lifespan data from Durham and colleagues [36], we detect no correlation between lifespan and climbing index in females (Pearson's product-moment correlation $r = 0.092$, $p = 0.6173$). This finding is confirmed using the lifespan data from Ivanov and colleagues [37] (Pearson's product-moment correlation $r = -0.065$, $p = 0.7202$). Similarly, when we investigate the relationship between animal weight and climbing ability (data from [32]), we find no significant correlation in either sex, in control or exercise-treated animals (Pearson's product-moment correlation, $p > 0.4$). These findings suggest that high physical fitness, as measured by a climbing index, is not linked to either lifespan or animal weight in these Drosophila strains. One possible explanation is that the climbing index measures a very short, limited burst of activity upon stimulation, which might not reflect the overall physical condition of an animal.

## 2.3. Excise impacts on climbing activity

Given the strong correlation between climbing indexes in exercise-treated and control animals, we next examined how climbing ability differed between exercise-treated and control animals. Figure 3 shows the average difference in climbing index between exercised and control animals of the same genotype and sex. Similar to what is observed in exercise studies in other systems including humans, the responses to exercise vary widely and include both positive responses, negative responses and a lack of response. In females, line 317 has the strongest decrease in climbing index with an approximately 10% decrease, while line 852 shows the strongest improvement in climbing index with exercise, with an approximately 20% increase. In males, line 852 shows the strongest decline in climbing index with a 13% decrease, while line 730 shows the strongest improvement with a 16% increase. Overall, females tend to show an increase in climbing index more often than males (18 versus 13 strains). Investigating further how males and females from the same DGRP line respond to exercise, there is no significant correlation between the change in climbing index in males versus females, neither for the raw change in climbing index, nor for the % change (Pearson's product-moment correlation $r = -0.126$, $p = 0.4842$ raw data; $r = -0.064$, $p = 0.7219$ per cent change). Together, the data suggest that how animals respond to exercise in terms of climbing ability is strongly dependent on the specific sex/genotype combination. In addition, these data confirm that exercise non-responders occur in the Drosophila model as well as in human populations.

Next, we investigated whether there was a relationship between the change in climbing ability with exercise and the change in activity levels that we see with an exercise treatment. If there is a simple relationship between exercise intensity and change in physical fitness, one would expect the animals that show the largest increases in activity with the exercise treatment to also show large improvements in climbing activity. In males, there is no significant correlation between activity change and climbing index change with exercise (figure 4; Pearson's product-moment correlation

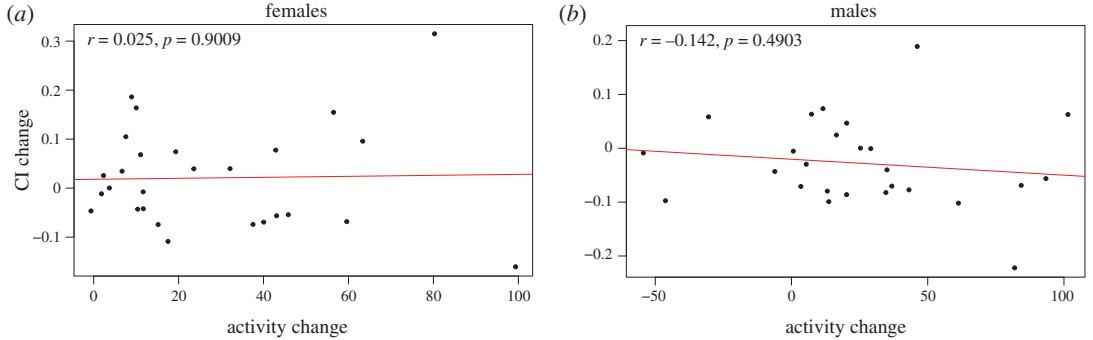

**Figure 4.** There is no relationship between the exercise-induced change in climbing ability and the exercise-induced change in activity level. The change in climbing index (CI, $CI_{exercised} - CI_{control}$) is plotted on the Y-axis against the exercise-induced change in activity level (activity with rotation − activity without rotation; average number of beam crossings recorded by the activity monitor in a 5 min interval for 10 flies) on the X-axis for each of the DGRP strains included in this study. (*a*) Data from females. (*b*) Data from males.

$r = -0.142$, $p = 0.4903$). Similarly, in females, which respond to rotational exercise stimulation with a larger increase in activity than males, we also see no significant correlation between the activity change and climbing index change with exercise (Pearson's product-moment correlation $r = 0.025$, $p = 0.9009$). Surprisingly, this finding suggests that there is no clear relationship between the amount of exercise activity and the change in climbing ability or fitness as measured by a change in climbing index.

## 2.4. Genes linked to climbing ability

To determine if the data collected might be suitable for a GWAS to discover the molecular pathways important for climbing ability and the response to exercise, we calculated relevant quantitative genetics statistics (electronic supplementary material, table S2). Importantly, we found that heritability for climbing ability is high both in control and exercised animals (0.610 and 0.638). This level of heritability suggests that a GWAS using this subset of DGRP strains might be able to identify molecular pathways impacting climbing ability.

First, we carried out GWASs with the climbing index data from control animals. We partitioned our data into two datasets, one based on images 1,3 and 5, one based on images 2, 4 and 6, which serve as validation studies for each other. The GWASs investigated genetic variation linked to climbing ability in the combined data from both sexes, the data separated by sex and the differences between the two sexes. In the data from images 1, 3 and 5, we identified 33 genetic variants in 23 genes, while we identified 112 genetic variants associated with 58 genes using the data from images 2, 4 and 6 (electronic supplementary material, table S3). Of the 33 genetic variants identified from the analysis of images 1, 3 and 5, 13 are linked to climbing ability in females, 11 are linked to climbing ability in males, eight variants are linked to climbing ability in the combined sex data and three variants are linked to the sex difference in climbing ability (electronic supplementary material, figure S3). Of the 112 genetic variants identified from the analysis of images 2, 4 and 6, 52 are linked to climbing ability in females, nine variants are linked to climbing ability in males, 10 variants are linked to climbing ability in the combined sex data (three overlapping with males, *dpr6*, *α-Man-IIb*, *Srp54 k*; 2 overlapping with females, intergenic and *CG33543*), and 45 variants are linked to the sex difference in climbing ability. In both datasets, there is little overlap between the genes contributing to climbing ability in males and females, indicating that different molecular pathways are important in the two sexes. Comparing the results from the analysis of the two image sets, we find that there are eight genes detected as candidate genes in both analyses, which is significantly more than expected by chance (hypergeometric test, $p = 5.495999 \times 10^{-15}$). These shared genes should be considered the strongest candidate genes, while the genes detected by only one analysis should be further investigated to gather additional support for a potential role in controlling climbing ability. These genes are *CG33125* (F), *CNMaR* (M\*/diff), *CG9498* (M), *dpr6* (F\*/M/combined), *Hrb27C* (F), *numb* (combined), *Srp54 k* (M/combined) and *unc-13-4A* (combined) [\* only significant in one image set], and they encode a diverse set of proteins. *CG33125* and *CG9498* are proteins of unknown function [38]. *CNMaR* encodes a G-protein coupled receptor for the CNMa neuropeptide [38,39]. *dpr6* was discovered in a genetic screen due to defective proboscis extension response [38,40]. *Hrb27C* encodes a nuclear

ribonucleoprotein, and mutants are described to have a small body and various neuroanatomical defects [38,41,42]. *numb* encodes an inhibitor of Notch signalling, and mutants are flightless (overexpression) and exhibit muscle defects [38,43–45]. *Srp54 k* encodes a signal recognition particle [38], and *unc-13-4A* mutations are associated with pain response defects [38,46]. Thus, the GWASs focused on climbing ability in control animals have identified a set of diverse candidate genes for further investigation.

Next, we carried out GWASs with the climbing data from exercise-treated animals. Again, we partitioned the data into two datasets (images 1,3, 5 versus 2, 4, 6) and set up GWASs using the combined data from both sexes, the data separated by sex and the differences between the two sexes. In the data from images 1, 3 and 5, we identified 45 genetic variants in 20 genes, while we identified 96 genetic variants associated with 53 genes using the data from images 2, 4 and 6 (electronic supplementary material, table S3). Among the 45 variants identified from the analysis of images 1,3 and 5, 10 were identified linked to climbing ability in exercised female animals, 24 were linked to climbing ability in exercised male animals (one overlapping with females, *dpr6*), 17 variants were identified in the combined sex data (12 overlapping with males; three overlapping with females, *dpr6*, *numb*, *CG3927*), and two variants were linked to the difference between males and females (electronic supplementary material, figure S3). Among the 96 variants identified using the data from images 2, 4 and 6, 43 variants were identified using the data from females, 23 using the data from males (2 overlapping with females, *lim3* and *dpr6*), 13 using the combined data from both sexes (11 overlapping with males, three overlapping with females, *lim3*, *numb*, *dpr6*) and five variants were associated with the difference in climbing ability between the sexes (electronic supplementary material, figure S3). Again, mostly distinct gene sets contribute to climbing ability in the two sexes. Comparing the results from both image sets, we find that there are six genes identified as candidate genes in both analyses, and that this overlap between the two datasets is significantly higher than expected by chance (hypergeometric test, $p = 1.974103 \times 10^{-11}$). These six genes are the strongest candidate genes controlling climbing ability after exercise, while the genes detected by only one analysis should be further investigated to gather additional support for a contribution to this phenotype. The genes are *CG3927* (F/combined*), *CG9498* (M), *dpr6* (F/M/combined), *Lim3* (F*/M*/combined), *numb* (F/combined) and *Tie* (M/combined) [* only significant in one image set]. The functions of *dpr6*, *numb* and *CG9498* have been discussed in the previous paragraph. CG3927 is a protein of unknown function [38]. Lim3 is a transcription factor that specifies neuronal cell identity, and animals lacking this protein have neuroanatomical and locomotion defects [38,47,48]. Tie is a transmembrane receptor tyrosine kinase [38,49]. Together, the results from the GWASs based on the climbing ability of exercised animals suggest that there are diverse types of genes contributing to this phenotype; many of them are different from the genes identified in the GWASs based on the data from control animals. However, there are three genes, *CG9498*, *dpr6* and *numb*, that are identified in all analyses and thus appear to be the strongest candidate genes linked to climbing ability in general.

## 2.5. Gene ontology analysis

To better understand the groups of genes linked by GWAS to climbing ability, we carried out gene ontology (GO) analysis. Specifically, we used the candidate genes detected in both the GWASs from control and exercised animals, excluding the genes linked to the sex difference, as these genes tended to show no overlap with the other categories. Analysing the resulting gene list with the PANTHER classification system [50,51], no GO terms for 'molecular function' were identified as over-represented. Among the GO terms for 'cellular component', the term 'integral component of plasma membrane' as well as some related terms was significantly enriched in the candidate gene set ( $p = 0.037$, Fisher's Exact test, FDR correction; electronic supplementary material, table S4). The GO term analysis focused on 'biological process' yielded the most interesting results (figure 5; electronic supplementary material, table S4). In total, 90 'biological process' GO terms were found to be significantly enriched in the gene set identified by the GWASs for climbing activity. Grouping these GO terms into related sets reveals that the majority of GO terms fall into three categories: 'pattern specification process', 'regulation of multicellular organismal process' and 'cell–cell signalling'. Two of the strongest candidate genes, *dpr6* and *numb*, both have functions in cell–cell signalling, supporting the importance of this specific GO term. Overall, the GO analyses suggest that, not surprisingly, climbing ability depends on genes that mediate cell-to-cell signalling, likely between several tissues, and on the proper formation of these tissues and communication pathways.

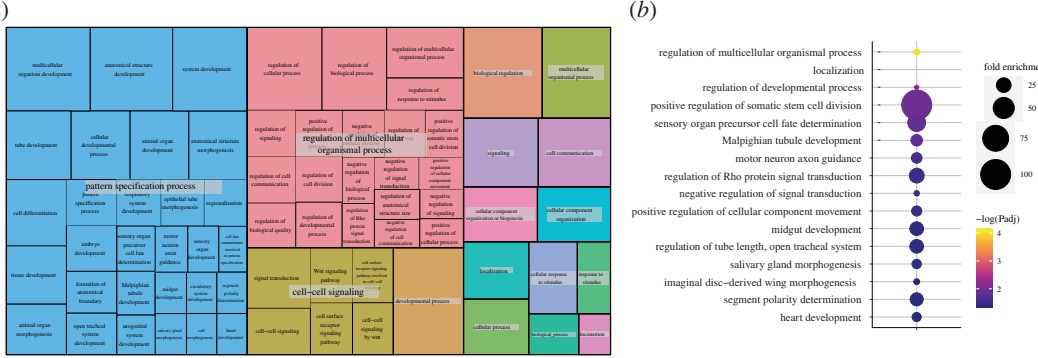

**Figure 5.** Cell–cell signalling and tissue patterning genes contribute to the variation in climbing ability. (*a*) Visualization of GO terms associated with climbing ability using REVIGO. Similar terms are clustered and reduced to higher level terms for easier viewing. (*b*) Detailed view of the top GO terms with fold enrichment shown by the size of the circle and the *p*-value shown by colour. For additional details, see electronic supplementary material, table S4.

## 2.6. Genes linked to exercise-induced change in climbing ability

While our GWASs for climbing ability in control and exercised animals are adequately powered to detect candidate SNPs ([52] implemented by https://github.com/kaustubhad/gwas-power.git), due to the lack of replicates for the 'change in climbing ability with exercise' phenotype, this part of the study is insufficiently powered to run two independent analyses as we have done with the earlier datasets. Thus, we decided to combine data from all images into a single dataset to carry out two separate analyses: (i) a targeted analysis to determine whether the strong candidate genes detected for climbing ability also impact the change in climbing ability in response to exercise and (ii) an exploratory GWAS. In the targeted analysis, comparing the response in climbing ability after exercise for lines with different variants in *dpr6*, *numb* and *CG9498* does not reveal an impact of these variants (*t*-test, *p* > 0.05). In the exploratory GWAS, we identify 54 genetic variants in 34 different genes. Nine variants were detected using the female data only, 13 were detected using the male data only, six were detected using the combined sex data, and 26 variants were detected as linked to the difference between the two sexes. Of note among these variants is one in *Hrb27C*, which was also detected in the GWAS of climbing ability in control animals, as well as a variant in *Nckx30C*, a calcium transporter that functions in calcium homeostasis during signalling, that was also identified in one of the GWAS of climbing ability in exercised animals. Given the small gene set, GO analysis did not reveal any enriched categories. Overall, the variants identified in this exploratory analysis for the genetic basis of the climbing ability response to exercise show little overlap with the candidate genes identified as associated with climbing ability alone, suggesting that different types of pathways are involved.

## 3. Discussion

Drosophila strains from the DGRP show a large amount of variation in climbing ability, both in control animals and animals that received a 5-day rotational exercise treatment. This variability is strongly dependent on sex and genotype, and genome-wide association studies implicate genes in cell–cell signalling pathways and multi-organ developmental processes in the control of climbing ability. *dpr6*, *numb* and *CG9498* were identified as strong candidate genes for climbing ability, and they provide promising avenues for further study. We documented extensive variability in the response to a 5-day rotational exercise treatment in terms of change in climbing ability, with the DGRP strains examined almost equally divided between strains that show an increase in climbing ability and those showing no change or decreased climbing ability. Thus, the Drosophila system replicates the existence of exercise non-responders often observed in human exercise studies. While the exploratory GWAS for exercise response in climbing ability revealed no enriched GO terms, the results indicate that the genes involved in this exercise-induced climbing ability change differ from those controlling climbing ability in general. Together, the results presented here demonstrate that climbing ability and its response to an exercise treatment are highly variable, dependent on sex and genotype, and impacted by diverse gene sets.

While other studies have investigated activity phenotypes including climbing ability in the DGRP strain collection, ours is the first study to do so in the context of an exercise treatment. Lavoy et al. [53] measured climbing ability in the DGRP in the context of Parkinson's disease (PD), crossing 148 DGRP lines to the *LRRK2 G2019S* expressing Drosophila PD model. Focusing on female flies, they find that climbing ability is highly dependent on genotype, in agreement with our results. However, if we compare the climbing ability measured in the homozygous DGRP strains here to the data collected by Lavoy in the PD model at two weeks of age, we find only a weak, non-significant correlation ($r = 0.203$, $p = 0.3193$, Pearson's product-moment correlation). This result suggests not surprisingly that climbing ability is impacted by both dominant and recessive genomic features and that climbing ability in the PD model is quite different from that in the wild-type situation. Interestingly, while the climbing abilities from their control cross with flies carrying a non-PD-associated *LRRK2* transgene are not reported, the GWAS candidates for climbing in the control cross are reported, and the top hit is *dpr6*, which is one of our top three candidates here as well. In addition, similar to our study, the results from the Lavoy study also suggest that the genes controlling climbing ability with and without a treatment can be very different, showing no overlap in their study [53]. The findings from both the Lavoy study and the work presented here thus suggest that climbing ability is modified by different genetic pathways depending on the treatment applied.

Another study of climbing ability in Drosophila focused on the response to a treatment with paraquat, a herbicide that is used to induce oxidative stress in Drosophila studies [54]. Lovejoy and colleagues investigated climbing ability in the DGRP comparing paraquat-treated animals to matched control animals, with a focus on sexual dimorphism. Similar to our results with the impact of exercise, Lovejoy and colleagues find strong impacts of sex and genotype on the change in climbing ability upon paraquat treatment. They also find that the impact of paraquat varies widely, with most animals showing a decrease in climbing, but a significant subset showing no change or increased climbing [54]. This wide range in treatment responses is reminiscent of what we see with the exercise treatment, and what was reported by Lavoy et al. [53] in their PD model. While Lovejoy and colleagues only carried out a GWAS analysis on the response to paraquat treatment, the raw data from the control animals were available in the electronic supplementary material. A GWAS with the same method used in the study presented here reveals a list of 48 candidate genes but no overlap with the candidate list from this study (electronic supplementary material, table S6). The reason for this lack of overlap is possibly in the different DGRP lines that were used or in the handling of the animals. Our animals were older than the animals used by Lovejoy and colleagues (9–11 days versus 4–5 days post eclosion). In addition, the protocol by Lovejoy and colleagues involved a 4 h starvation period followed by 24 h on a sucrose only diet, while our animals were maintained on standard cornmeal media throughout their lives. However, while there is no overlap in the specific candidate genes identified between the two studies, genes linked to the pattern formation (cell fate determination' and dorsal/ventral pattern formation) are enriched here as well based on GO term analysis, as are genes involved in axon guidance and chemotaxis (electronic supplementary material, table S6). Taken together, these results suggest that climbing ability is highly plastic, changing in response to a variety of environmental factors, and linked to different genetic pathways depending on the specific environment and experimental treatment.

Given the positive correlation we detected between the startle response measured by Mackay et al. [33] and the climbing ability measured here, we also looked at the overlap between genes linked to the startle response and the candidate genes identified here for climbing ability. Startle response assays measure the fly's locomotor activity to a disturbance in two dimensions, typically in a small Petri dish, recording the fraction of time an animal is active in a pre-determined time interval (e.g. 30 s post-disturbance). Among the candidate genes identified by Mackay et al. [33] is *dpr6*, which is linked to climbing ability in this study, as well as two other genes that are identified in some but not all of our analyses (*pgant2* and *CG5888*). Similarly, candidate genes for startle response identified in an earlier study from the Mackay laboratory, also show overlap with the less well-supported candidate genes in our study (*nrm*, *nrv2*, *Rheb*) [55], suggesting that they might be important for locomotor behaviour in general. A re-analysis of the data from Mackay and colleagues, taking into account transcript variation in addition to genetic variation, identified additional candidate genes linked to startle response that overlap with those detected here for climbing ability (*Pura*, *CG33158*) [56]. The correlation between startle response behaviour and climbing assay measures as well as the shared candidate genes suggests that there are at least some common pathways that control both phenotypes, likely those pathways linked to the basic ability to respond to a stimulus (disturbance in the case of the startle assay, fall to the bottom of the vial in the case of a climbing assay) and overall walking ability.

One unexpected result from this study is the lack of correlation between climbing ability and baseline activity levels or the amount of exercise-induced activity and the exercise-induced change in climbing activity. We expected to see animals with higher baseline activity levels perform better in climbing assays than animals with low baseline activity levels. Similarly, we expected that animals that show larger increases in activity levels with exercise would show the most improvement in climbing ability after an exercise treatment. The lack of correlation between basal activity levels and climbing ability as measured by the RING assay suggests that basal activity levels are a poor measurement of a strain's ability under the RING assay conditions. Rather, it suggests that the basal activity levels might reflect a preferred activity level rather than ability and physical fitness or body condition. It is interesting; however, that there is also no correlation between the climbing ability measured and the exercise-induced activity measures, suggesting that this higher level of activity is not necessarily reflected in the animals' climbing ability. This finding also raises the question of whether activity measures might not be tied more closely to endurance measures of fitness rather than the short 'sprint' that is measured with the climbing assay. This interpretation is supported by the fact that the climbing index measures show a significant positive correlation with startle response measures.

Another possible reason for the lack of correlation between exercised-induced activity levels and an exercise-induced change in climbing ability might be the nature of the specific exercise treatment chosen in the study presented here. It is possible that the relatively short, 5-day exercise treatment is insufficient to illicit a response in many of the lines. However, given the reported lifespans of the DGRP lines (55–75 days), 5 days correspond to a significant portion of their lifespan and is comparable to the lengths of treatments typically studied in mammals. In addition, it is possible that exercise intensity needs to be considered as well as the length of the exercise treatment. However, accurately measuring exercise intensity in Drosophila is difficult and likely will require the development of new methods to assay respiration rates during exercise treatments.

Finally, the lack of correlation between exercise-induced activity change and change in climbing ability suggests that exercise responses do not simply reflect the amount of additional activity engaged in by the subjects. Instead, it appears that there are nonlinear relationships between exercise amount and exercise response, and that potentially other parameters need to be considered. Given the complexity of animal behaviour, it is likely that some Drosophila strains compensate for exercise by lowering activity levels during the rest of the day. They might respond to exercise with a change in food intake, either increase or decrease. Animals also might have different trajectories for how they recover from fatigue due to exercise, resulting in different performance levels at the time point when the climbing ability was measured. Based on the data available, it seems likely that there is an optimal exercise amount, type and pattern for each genotype-sex-age combination and that exercise either below or above this optimal level leads to poor outcomes. Research in model systems like *D. melanogaster* and *C. elegans* will be essential to explore the large variable space that needs to be tested to understand what factors determine optimal exercise treatments.

While climbing ability is an established measure for physical fitness in Drosophila, we detect no correlation between climbing ability and lifespan in our study. This finding is unexpected, as typically, it is believed that physical fitness is associated with longer life. Our finding is in agreement with the work by Wilson and colleagues, who investigated the impact of dietary restriction on age-related changes in climbing ability and on lifespan [57]. They find that changes in climbing ability with age are unrelated to lifespan, showing no significant correlation [57]. This finding is similar to the lack of correlation we find for climbing ability and lifespan. Looking at other species, in *C. elegans*, swim exercise treatments increase median lifespan, but not maximum lifespan [18]. The swim exercise also positively impacts other measures of healthspan such as learning and maximal respiration, but the impact of exercise and any potential benefit is strongly dependent on the specific exercise treatment [18]. Thus, the findings in two model systems highlight the complex relationship between exercise, physical fitness, age-related physical decline and lifespan, and suggest that the relationship between physical fitness and lifespan is more complicated than previously thought. This insight is underscored by the fact that in humans, exercise impacts on lifespan are highly variable as well, and that optimal exercise levels, type, frequency etc. are currently unknown and likely differ between individuals and change with age.

## 3.1. Conclusion

In summary, this study provides important insights into the mechanisms controlling physical fitness as measured by climbing ability in Drosophila. It confirms that climbing ability in Drosophila is highly sex-

and genotype-specific and demonstrates that climbing ability can be altered by the application of a rotational exercise treatment. The GWASs identify *dpr6*, *numb* and *CG9498* as clear candidate genes and point to cell–cell signalling networks as a promising area of further study for their role in exercise response. The results presented also highlight the need for additional studies to clearly delineate the impact of exercise on response phenotypes like climbing ability and to define optimal exercise regimes. To improve exercise treatments for individuals, it is essential that we understand how genotype, sex and age interact to determine if a response to a specific exercise treatment is positive, negative or non-existent. Solving this question is a significant undertaking, but studies in cost-efficient model systems like *D. melanogaster* and *C. elegans* can provide information about candidate genes and pathways to augment studies in mammalian models and humans.

# 4. Material and methods

## 4.1. Fly husbandry

All flies used in this study were from the DGRP [33,58] and are listed in electronic supplementary material, table S1. The flies used in this study were reared in a strictly controlled environment (incubator settings: 25°C, 70% humidity and a 12 h/12 h light/dark cycle [light on 7 : 00–19 : 00 PM]). Flies were raised in vials on media consisting primarily of cornmeal, agar, molasses and yeast, with propionic acid and Tegosept added as antifungal agents [24].

## 4.2. Exercise regime

Exercise treatments were administered using the TreadWheel following the protocol used in 'Study B' from The University of Alabama at Birmingham [24]. Typically, four to five genotypes were processed per week. Briefly, virgin animals were collected and aged to 3–5 days. Groups of 20 animals were moved into vials with 1 inch of food using 100 animals (5 vials) per sex/genotype/treatment combination. While treated animals exercised for 2 h each day for 5 days, controls were handled identically and placed on a platform attached to the TreadWheel for the animals to experience the same vibration and noise as the exercise-treated animals. Flies were loaded onto the TreadWheel at 11 A.M. each day, followed by a 1 h acclimatization period. At noon, rotation was initiated at four rotations-per-minute, and the flies were exercised for 2 h until 14.00. Following each exercise, the vials containing flies were unloaded from the TreadWheel and returned to the incubator under standard conditions until the next day. After the last exercise treatment, the animals were kept in the incubator overnight before physical fitness was assayed.

## 4.3. Rapid iterative negative geotaxis assay

A protocol adapted from the RING was used to assay climbing ability as a measure of physical fitness [59]. On the day following the last bout of exercise, starting at approximately 13.00, flies were anesthetized with $CO_2$ and loaded into a custom-made RING assay apparatus in groups of 20 animals separated by sex/genotype/treatment combination. As $CO_2$ anaesthesia can impact animal behaviour, exposure time was kept to a minimum (less than 5 min), and all animals, treated and untreated, experienced the same anaesthesia [60–62]. This RING assay apparatus consists of glass vials (length: 5 inches) that have been glued to a standard vial rack with weather-stripping foam tape at the bottom to ameliorate the effects of repeatedly tapping the apparatus to a solid surface. Foam plugs were used to close the vials. Prior to beginning the experiment each day, a camera was set to a 2 s timer. The RING assay apparatus was tapped down five times to ensure that all flies were at the bottom of the vials. On the fifth tap, a second researcher operating the camera started the timer, allowing the flies to climb for two seconds prior to the picture being taken. Flies were then given a 1 min rest period, and the process was repeated to capture a total of six pictures for each set of animals.

The images were analysed by dividing the vial height into four equal quarters using an image processing software. The number of flies in each quarter was counted for each picture by a researcher blinded to the sample identity. A climbing index was calculated by assigning each quarter a point value, with quarters located higher up in the vial receiving more points (top quarter—four points; top middle quarter—three points; bottom middle quarter—two points; bottom quarter—one point). All

scores from one vial were added and then divided by the number of flies in the vial (typically 20) to result in a climbing index. All raw data can be found in the electronic supplementary material, table S1.

## 4.4. Genome-wide association studies

GWASs were carried out using the DGRP2 webtool (http://dgrp2.gnets.ncsu.edu/), which automatically carries out GWASs for data collected from the DRGP strains [33,58]. The webtool executes separate analyses of data from males and females, an analysis of the combined data, as well as an analysis of the sex differences. $p$-values are reported for each genetic variant for a mixed effects model as well as a simple regression model. The models implemented by the DGRP2 webtool adjust for the presence/absence of Wolbachia and five major inversions present within this strain collection [In(2L)t, In(2R)NS, In(3R)K, In(3R)P and In(3R)Mo]. Input data for the DGRP webtool are the group means based on genotype/sex/treatment combinations calculated from the raw data in the electronic supplementary material, table S1. We used a nominal $p$-value cutoff of $10^{-5}$ to identify candidate genes for further consideration [33,63–68], as this approach has been successful in other studies of the DGRP and is supported by inspection of qq-plots.

To investigate climbing ability in control and exercised animals (GWAS for climbing index), the data were divided into two replicate datasets, one based on images 1, 3 and 5, one based on images 2, 4 and 6. The two datasets were analysed separately, and the results were compared to identify the most promising candidate genes.

## 4.5. Gene ontology term analysis

GO term analysis was carried out using the PANTHER (Protein ANalysis THrough Evolutionary Relationships; version 16.0) Classification System [50,51]. A false discovery rate (FDR; $p < 0.05$) was used for multiple testing correction in the overrepresentation test (Fisher's Exact test). Revigo was used to visualize the significant GO terms [69].

## 4.6. Statistical analysis

Statistical analyses were performed using R [70]. For the linear mixed effects analysis, we used the 'lme4' package [71]. Sex, genotype and treatment were coded as fixed effects, while position in the assay rigg and vial were coded as a random effect (random intercepts). 'Picture' was not included in the final model, as it did not improve the model (AIC −1773.3 versus −1686.0, $p < 2.2 \times 10^{-16}$). $p$-values for specific effects were generated by likelihood ratio tests comparing the full model to a model lacking the effect being tested. While a model with sex, genotype treatment and all their interactions (along with position and vial) is close to optimal (AIC: −1773.3), a model with only the sex : genotype and sex : treatment interaction is a better fit (AIC: −1774.1, $p = 3.77 \times 10^{-6}$). Using a 'drop-1-out' approach, we find that removing the sex : treatment interaction results in a slightly better fit (AIC: −1774.7), but not significantly so ($p = 0.2481$). Plotting residuals did not reveal any obvious deviations from normality.

Quantitative genetics parameters such as variances components and heritability (electronic supplementary material, table S2) were calculated using the 'VCA' package [72].

Data accessibility. All data are included in the electronic supplementary material of the submission. The .zip file 'Unprocessed_data_scripts.zip' contains scripts and pre-processed data files for the reviewers. Analyses and figures were prepared in R.

The data are provided in the electronic supplementary material [73].

Authors' contributions. L.P.W. participated in the design of the study, carried out the data collection and preliminary data analyses, and critically revised the manuscript. N.C.R. conceived of the study, designed the study, coordinated the study, carried out the final data analyses and drafted the manuscript.

Competing interests. We declare we have no competing interests.

Funding. This study uses stock provied by the Bloomington Stock center, which is funded by NIH (grant no. P40OD018537 (To the Bloomington Drosophila Stock C).

Acknowledgements. We would like to thank the following undergraduate students that were involved in evaluating the images for the RING assay: Stephanie Moore, Sarah K. Sims. We would also like to thank Cameron Gordon and Michael Azar for helping with fly husbandry and the fly exercise treatments. We thank Dr L. Reed (University of Alabama) for providing many of the DGRP stocks. Stocks obtained from the Bloomington Drosophila Stock Center (NIH P40OD018537) were used in this study.

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
