## [Peer Review File · Royal Society Open Science]

Review History

RSOS-211275.R0 (Original submission)

Review form: Reviewer 1 (Robert Wessells)

Is the manuscript scientifically sound in its present form?

Yes

Are the interpretations and conclusions justified by the results?

No

Is the language acceptable?

Yes

Do you have any ethical concerns with this paper?

No

Have you any concerns about statistical analyses in this paper?

No

Recommendation?

Accept with minor revision (please list in comments)

Comments to the Author(s)

Manuscript presents comparison of climbing speed across various genotypes, before and after a 5-day, low-intensity exercise program. Some overlap in conclusions exists between the manuscript and previous work by the authors and others, but work is still of interest and extensive dataset will be useful to the field. A few clarifications of interpretation via text revision would increase enthusiasm for this very nice, and thoroughly analyzed, work.

1. Since baseline activity and REQS (distance travelled during exercise) are both analyzed for correlation with RING (climbing speed), would avoid confusion to standardize description of these different measures to avoid terms like "climbing ability."
2. Would be worth discussing whether CO₂ anesthesia prior to exercise may influence efficacy as it has been reported that CO₂ treatment can have short and long term effects on various aspects of mobility.
3. Authors divide their 6 pictures of each set and mention that the two subsets can be used as independent validation. However, the two sets identified different potential genes. How should the reader interpret this in the context of validation?
4. Lack of relationship between REQS during exercise and improvement in climbing index following exercise treatment is a very interesting aspect that should be discussed more fully in the context of how this compares to other animals, including humans. Could this be related to the intensity of the exercise? For example, high-intensity interval training is a growing field of study in humans in which length of activity appears to be less important for adaptations than the intensity of the activity.
5. Likewise, would be useful for authors to address whether the short timing of training (5 days) may have an impact on whether the exercise effect can overcome the sex and/or genotype effects?

Decision letter (RSOS-211275.R0)

Dear Dr Riddle

On behalf of the Editors, we are pleased to inform you that your Manuscript RSOS-211275 "Exercise-induced changes in climbing performance" has been accepted for publication in Royal Society Open Science subject to minor revision in accordance with the referee's reports. Please find the referee's comments along with any feedback from the Editors below my signature.

Please submit your revised manuscript and required files (see below) no later than 7 days from today's (ie 05-Oct-2021) date. Note: the ScholarOne system will 'lock' if submission of the revision

is attempted 7 or more days after the deadline. If you do not think you will be able to meet this deadline please contact the editorial office immediately.

on behalf of Professor Steve Brown (Subject Editor)
openscience@royalsociety.org

Reviewer comments to Author:

Reviewer: 1

Comments to the Author(s)

Manuscript presents comparison of climbing speed across various genotypes, before and after a 5-day, low-intensity exercise program. Some overlap in conclusions exists between the manuscript and previous work by the authors and others, but work is still of interest and extensive dataset will be useful to the field. A few clarifications of interpretation via text revision would increase enthusiasm for this very nice, and thoroughly analyzed, work.

1. Since baseline activity and REQS (distance travelled during exercise) are both analyzed for correlation with RING (climbing speed), would avoid confusion to standardize description of these different measures to avoid terms like "climbing ability."
2. Would be worth discussing whether CO2 anesthesia prior to exercise may influence efficacy as it has been reported that CO2 treatment can have short and long term effects on various aspects of mobility.
3. Authors divide their 6 pictures of each set and mention that the two subsets can be used as independent validation. However, the two sets identified different potential genes. How should the reader interpret this in the context of validation?
4. Lack of relationship between REQS during exercise and improvement in climbing index following exercise treatment is a very interesting aspect that should be discussed more fully in the context of how this compares to other animals, including humans. Could this be related to the intensity of the exercise? For example, high-intensity interval training is a growing field of study in humans in which length of activity appears to be less important for adaptations than the intensity of the activity.
5. Likewise, would be useful for authors to address whether the short timing of training (5 days) may have an impact on whether the exercise effect can overcome the sex and/or genotype effects?

===PREPARING YOUR MANUSCRIPT===

===PREPARING YOUR REVISION IN SCHOLARONE===

-- If you have uploaded ESM files, please ensure you follow the guidance at <https://royalsociety.org/journals/authors/author-guidelines/#supplementary-material> to include a suitable title and informative caption. An example of appropriate titling and captioning may be found at [https://figshare.com/articles/Table_S2_from_Is_there_a_trade-off_between_peak_performance_and_performance_breadth_across_temperatures_for_aerobic_sc ope_in_teleost_fishes_/3843624](https://figshare.com/articles/Table_S2_from_Is_there_a_trade-off_between_peak_performance_and_performance_breadth_across_temperatures_for_aerobic_scope_in_teleost_fishes_/3843624).

Author's Response to Decision Letter for (RSOS-211275.R0)

See Appendix A.

Decision letter (RSOS-211275.R1)

Dear Dr Riddle,

I am pleased to inform you that your manuscript entitled "Exercise-induced changes in climbing performance" is now accepted for publication in Royal Society Open Science.

on behalf of Professor Steve Brown (Subject Editor)
openscience@royalsociety.org

Appendix A

Response to Reviewers

We very much appreciate the thoughtful comments provided by the reviewer. Below is a point-by-point discussion of how we have addressed the various concerns raised.

“Since baseline activity and REQS (distance travelled during exercise) are both analyzed for correlation with RING (climbing speed), would avoid confusion to standardize description of these different measures to avoid terms like “climbing ability.”

We appreciate this suggestion from the reviewer and have edited to manuscript text to ensure consistency in the terms used.

“Would be worth discussing whether CO2 anesthesia prior to exercise may influence efficacy as it has been reported that CO2 treatment can have short and long term effects on various aspects of mobility.”

We have added information about the impact of CO2 treatment on fly behavior as well as the appropriate references to section 4.3.

“Authors divide their 6 pictures of each set and mention that the two subsets can be used as independent validation. However, the two sets identified different potential genes. How should the reader interpret this in the context of validation?”

It is very common for GWAS validation studies to identify different sets of SNPs and genes linked to a given phenotype. Of most interest are the genes that are identified in multiple analyses, and these genes should be considered the strongest candidate gene for follow-up studies. Genes identified only in one study should be considered weak candidate genes, unless other data, i.e. functional studies etc, that support their role in generating the phenotype understudy. We have added this information to the manuscript in section 2.4.

“Lack of relationship between REQS during exercise and improvement in climbing index following exercise treatment is a very interesting aspect that should be discussed more fully in the context of how this compares to other animals, including humans. Could this be related to the intensity of the exercise? For example, high-intensity interval training is a growing field of study in humans in which length of activity appears to be less important for adaptations than the intensity of the activity.”

“Likewise, would be useful for authors to address whether the short timing of training (5 days) may have an impact on whether the exercise effect can overcome the sex and/or genotype effects?”

We have added a paragraph to the discussion to clarify these points.